# Impact of Human Adenovirus 36 on Embryonated Chicken Eggs: Insights into Growth Mechanisms

**DOI:** 10.3390/ijms25042341

**Published:** 2024-02-16

**Authors:** Aleksandra Pogorzelska, Piotr Kuropka, Dominika Stygar, Katarzyna Michalczyk, Elżbieta Chełmecka, Jolanta Zalejska-Fiolka, Agata Kublicka, Paulina Janicka, Barbara Bażanów

**Affiliations:** 1Department of Pathology, Division of Microbiology, Faculty of Veterinary Medicine, Wroclaw University of Environmental and Life Sciences, 31 C. K. Norwida Street, 50-573 Wroclaw, Poland; aleksandra.pogorzelska@upwr.edu.pl (A.P.); agata.kublicka@upwr.edu.pl (A.K.); paulina.janicka@upwr.edu.pl (P.J.); 2Department of Biostructure and Animal Physiology, Division of Histology and Embryology, Faculty of Veterinary Medicine, Wroclaw University of Environmental and Life Sciences, C. K. Norwida 25, 50-635 Wroclaw, Poland; piotr.kuropka@upwr.edu.pl; 3Department of Physiology, Faculty of Medical Science in Zabrze, Medical University of Silesia, 19 Jordana Street, 40-808 Zabrze, Poland; dstygar@sum.edu.pl (D.S.); katarzyna.michalczyk@sum.edu.pl (K.M.); 4SLU University Animal Hospital, Swedish University of Agricultural Sciences, SE-750 07 Uppsala, Sweden; 5Department of Medical Statistic, Faculty of Pharmaceutical Sciences in Sosnowiec, Medical University of Silesia, 30 Ostrogórska Street, 40-200 Sosnowiec, Poland; echelmecka@sum.edu.pl; 6Department of Biochemistry, Faculty of Medical Sciences in Zabrze, Medical University of Silesia, 19 Jordana Street, 41-808 Zabrze, Poland; jzalejskafiolka@sum.edu.pl

**Keywords:** embryonated chicken eggs, human adenovirus 36, organogenesis

## Abstract

Human adenovirus 36 (HAdV-D36) is presently the sole virus identified to be associated with an elevated risk of obesity in both humans and animals. However, its impact on embryonated chicken eggs (ECEs) remains unexplored. This study endeavoured to examine the influence of HAdV-D36 on embryonic development by utilizing embryonated chicken eggs as a dynamic model. To simulate various infection routes, the allantoic cavity and the yolk sac of ECEs were inoculated with HAdV-D36. Subsequently, embryos from both the experimental (inoculated with virus) and control (inoculated with PBS) groups were weighed and subjected to daily histological examination. The daily embryo weights were assessed and compared between groups using the Shapiro–Wilk test. Histopathological changes in tissues were examined and compared between the tested and control groups to ascertain physiological alterations induced by the virus. Our study confirmed a significant increase in the body weight of ECEs. However, this phenomenon was not attributable to adipose tissue development; rather, it was characterized by an augmented number of cells in all observed tissues compared to control subjects. We posit that HAdV-D36 may impact developing organisms through mechanisms other than enhanced adipose tissue development. Specifically, our findings indicate an increased number of cells in all tissues, a phenomenon that occurs through an as-yet-unexplored pathway.

## 1. Introduction

Virus-associated obesity has gained interest in the scientific world in recent years. Viral infection and obesity, two seemingly unrelated events, come together in the case of infectious obesity. The term includes an increase in body weight associated with the presence of an infectious agent, e.g., virus or bacteria [1]. Pasarica et al. indicated four types of viruses whose presence in the host organism may cause obesity: canine distemper virus (CDV), Rous-associated virus number 7 (RAV-7), Borna disease virus (BDV), and adenoviruses, mainly avian adenovirus (SMAM-1) and human adenovirus 36 (HAdV-D36) [2]. However, only human adenovirus 36 has been confirmed to infect both humans and animals [3].

Human adenovirus 36 (HAdV-D36) belongs to the *Adenoviridae* family in group D. It is a non-enveloped virus with a stable, highly conserved genome consisting of double-stranded DNA (dsDNA) with a rate of genetic variations of 2.37 × 10^−6^ mutations/nucleotide/passage, reported by Nam et al. [4]. Adenoviruses cause mainly respiratory and digestive system diseases [5]. Research on adenoviruses associated with obesity has been conducted since 1990, when SMAM-1’s pro-obesity effects were first discovered in birds [6]. Further research of the *Adenoviridae* family, with particular emphasis on human adenovirus 36, showed that this virus significantly reduces the amount of triglycerides and cholesterol [7] in the blood and has a simultaneous adipogenic effect. The virus affects the differentiation of 3T3-L1 preadipocytes into adipocytes [8]. Interestingly, insulin plays an important, inductive role in adipocytogenesis [9]. Rats infected with HAdV-D36 present weight gain, greater insulin sensitivity, and increased glucose uptake. Studies on rats, adult chickens, and monkeys showed a 15–30% increase in body weight after HAdV-D36 infection [10]. Moreover, HAdV-D36 was easily transmitted between infected and uninfected animals via droplets. Adenovirus 36 causes obesity by simultaneously activating several physiological processes: activation of pre-adipocyte differentiation into adipocytes, increase in glucose uptake, decrease in leptin and norepinephrine levels, and increase in the MCP-1 level [3]. Worldwide studies on various populations have shown the presence of antibodies against adenovirus 36 in the majority of the studied subjects. Almgren et al., researching the Swedish population, noted the presence of HAdV-D36 in more than half of the patients and an increase in virus prevalence from 7% in 1992–1998 to 15–20% in 2002–2006 [11]. Zhou et al., studying 824 patients in China, estimated the overall seroprevalence of human adenovirus 36 at 49.8% [12]. Similar studies conducted worldwide present similar results, indicating the increasing potential role of HAdV-D36 involvement in the incidence of infectious obesity.

So far, the mechanisms of action have been studied and described only in in vitro models or adult model organisms. Since the metabolism of embryos is very distinct from that of adult organisms, more focused towards growth and differentiation, it may respond differently to the virus’s presence. Viral infection, like insulin, may possibly act as a glucose uptake inductor or growth factor [8,9].

In the case of HAdV-D36, no data on its influence on embryos exist, and the virus affects adult model organisms via a complex mechanism; thus, we believe that studies on developing organisms, such as embryonated chicken eggs (ECEs), can deliver a glimpse of new knowledge about human adenovirus 36’s biology, applicable in both veterinary and human medicine. The fertilized avian egg constitutes one of the longstanding research models employed in virology. The clean conditions within the egg, coupled with its provision of diverse tissues to which the virus can show tropism, render this research model versatile and well-suited for conducting preliminary studies. Numerous factors have a role in the pathophysiology of metabolic syndrome and obesity. Because of the comorbidities associated with pathological obesity, including diabetes, cardiovascular disease, hepatic steatosis, and non-alcoholic fatty liver disease (NAFLD), oxidative stress is considered a major factor in obesity and a serious health burden globally [13]. Reactive oxygen species (ROS), which are produced by oxidative stress, trigger both enzymatic and non-enzymatic antioxidant systems and neutralize them. For this reason, the effects of HAdV-D36 on selected oxidative stress markers in the analysed experimental model of obesity were tested in this study.

## 2. Results

### 2.1. ECE Weight Changes from the 6th to 20th Day of Development

Every day from the 6th to the 20th day of the incubation period, 28 ECEs from a total of 420 were examined. All ECEs were opened, and the embryos were weighed. Embryos inoculated with HAdV-D36 had a significantly different weight compared to control embryos (inoculated with PBS). The greatest differences between the HAdV-D36-infected and control embryos were found for those that were inoculated into the yolk. In this group, weight differences between the HAdV-D36-infected and control embryos ranged between 0.14 g and 1.09 g up to day 12 of development. In embryos older than 12 days, the greatest weight differences were noted for day 15 (3.51 g) and day 18 (5.41 g). In the case of embryos inoculated into the allantoic cavity, the differences were not as significant, ranging between 0.1 g (day 6 of development) and 3.33 g (day 20 of development). On days 6, 10, 17, and 19, the weight of control embryos was even higher than that of the infected ones.

These differences are not comparable to results from other studies investigating the effect of viruses on ECEs, because those viruses that affect foetus development have the opposite effect—they inhibit development, making the embryo weight lower compared to controls. Dwarfing of embryos is one of the effects with virus multiplication.

Detailed data from our investigations into the weights of embryos infected in the yolk sac and in the allantoic cavity are presented in Table 1.

An analysis of weight changes of the studied embryos showed that a polynomial relationship of the second degree describes the situation most accurately (Figure 1). We observed a significant change in weight gain occurring between days 12 and 13 (the area marked with a grey dashed line in Figure 1), followed by greater weight gains past day 12 in all study groups.

A further analysis of the data, including the observed shift in weight gains in older embryos, showed that the embryo development is better described by separate linear correlations: one for the period of day 6 to day 12, and a second for day 13 to day 20 (Figure 2).

For the earlier development (only the 6th to 12th days), all the obtained linear correlations were statistically significant (*p* < 0.001), and the regression coefficients showed a very strong positive correlation. We noticed that the β coefficients for HAdV-D36-infected ECEs were almost equal when comparing the ECEs inoculated into the allantoic cavity and into the yolk sac (0.6437 and 0.6483, respectively). In the case of the control ECEs inoculated into the yolk sac, we observed a much lower value of the β coefficient when compared to control ECEs inoculated into the allantoic cavity (0.4936 vs. 0.7429). For later ECE development (13th to 20th day), we obtained much higher slope coefficients (β) than for the early development for all ECE groups, indicating greater mean weight gains in the embryos’ later development. We observed the same tendency as for the earlier development—the ECEs infected with HAdV-D36 in the allantoic cavity and in the yolk sac had similar β coefficient values (3.7137 and 3.3257, respectively); however, surprisingly, the ECEs inoculated into the allantoic cavity presented a slightly stronger increase in weight. This tendency also occurred in the control ECEs; we observed greater weight gain in control ECEs inoculated into the allantoic cavity compared to those inoculated into the yolk sac.

### 2.2. Histopathological Examinations

No significant differences were discerned among the animals comprising the observed cohort. The sole factor exhibiting a marginal disparity between the groups was the embryo mortality rate, which exhibited an elevated incidence in the control group; however, it is imperative to interpret this occurrence as stochastic in nature. We observed changes in embryonic development as early as 48 h after HAdV-D36 infection. In general, we noted accelerated proliferation of whole-body cells. More cells, as well as an increase in the intercellular matrix volume, were observed in the dermis at the 7th day of incubation in chickens infected by HAdV-D36. Moreover, early connective tissue contained more thin collagen fibres and, therefore, was more optically dense than that in the control. Both the liver and the kidney presented accelerated growth and increased angiogenesis due to excessive proliferation.

On day 8, HAdV-D36-infected embryos presented increased diameter of their vertebral cartilages, dorsal muscles, liver, and kidneys. After observing the embryos for the first 4 days, it was possible to conclude that the HAdV-D36-infected embryos’ development was about 2 days more advanced than the development of the control embryos. On the 9th day of development (Figure 3) in the HAdV-D36 group, it was possible to notice the initial stages of vertebral cartilage hypertrophy and calcification, the formation of joints in the limbs, or the initial phases of eye tissue organisation.

The liver and kidneys were enlarged and contained more parenchymal cells when compared to sinusoid vessels filled with blood.

No increase in the number of adipocytes in the chicken body was observed. On the 10th day of HAd-VD36 infection, we observed the initial phases of respiratory tract development, including clear formation of the lungs, which is unique for this stage of development of birds, as normally they begin to develop later. On the 13th day of the HAdV-D36-infected embryos’ development, we observed a glandular period in the developing lungs and the first drops of fat appearing in the liver. We observed a constantly increasing size of all organs, with particular emphasis on the liver and kidneys, which grew significantly faster every day until day 20 of embryo development. However, on day 15 (Figure 4), we observed the first inflammatory changes in the liver, which were followed by extensive lymphocyte–macrophage infiltration to the liver parenchyma and the vicinity of blood vessels in the following days. On days 19 and 20 (Figure 5), we observed substantial liver inflammation with the initial stages of a very pronounced liver lipidosis.

### 2.3. Real-Time PCR

The analysis of real-time PCR results showed that the highest CQ (quantification cycle) concentration of HAdV-D36 genetic material in the embryos’ livers was found on day 20 and the lowest on day 17 for both tested groups (Figure 6 and Figure 7). Despite the low CQ value, viral DNA was present in each of the tested samples from HaDV-D36-infected embryos. The negative control showed no rise in CQ levels.

### 2.4. Antioxidant and Lipid Peroxidation Markers

We found statistically significant differences in CAT activity measured in the heart tissue of HAdV-D36-infected and control embryos (Table 2), with higher activity noted in HAdV-D36-infected embryos. We also found differences in SOD activities between the tested groups. Higher total SOD activity was observed in the study group. We observed statistically significant differences in MnSOD activity, with higher results noted for the infected group. Additionally, we noted statistically significant differences in CuZnSOD activity between the experimental and control groups. The study group presented higher CuZn SOD activity. As for GST activity, we also noted statistically significant differences between the HadV-D36-exposed and control groups, with higher results noted for the study group. We found no differences in the GPx (*p* = 0.542) and GR (*p* = 0.569) activities, nor in TOS levels (*p* = 0.826), between the two tested groups of ECEs. As for the tested lipid peroxidation marker, we observed a significantly higher MDA concentration in the treated group.

We found no differences in the CAT activity measured in the liver tissue of the two tested groups (*p* = 0.617). The total SOD activity in the liver tissue of the study group was significantly higher when compared to that of the control ECEs (Table 3). A similar effect was observed for CuZnSOD activity but not for MnSOD activity (*p* = 0.623). We observed higher GPx activity in the liver of the control group, while GR activity was lower. GST activity was at the same level in both tested groups (*p* = 0.258). The total oxidative status presented differences between the tested groups, with higher levels observed in the liver of the HAdV-D36-infected group. A similar effect was observed for MDA concentrations, with higher results observed in the study group.

## 3. Discussion

The influences of human adenovirus 36 on a host organism are relatively well described, although this knowledge still needs to be completed in a few areas. Nevertheless, the studies conducted thus far on animal models and humans have not incorporated organisms during the prenatal phase. Due to ethical reasons, no human studies have been carried out on embryos or foetuses. Our data suggest that HADV-D36’s effect on embryonated chicken eggs (ECEs) may result from a slightly different mechanism than the one described by Poterio et al., who described mechanisms of viral action encompassing processes such as adipogenic accumulation or adipocyte differentiation, which were not observed in our experiments [3]. Different types of adenoviruses show different abilities to reprogram host cells, one mechanism being the initiation of proliferative processes leading to oncogenesis. It is possible that HAdV-D36 induces an early oncogenesis pathway and later activates a different pathway leading to enhanced lipogenesis [14]. Due to the limited capacity of studies using the chicken embryo model, it was not possible to investigate other consequences of adenovirus 36 infection, such as heightened cellular glucose uptake and endocrine disruption.

Our measurements of the chicken embryos’ weight changes agree with the observations made by Pasarica et al., Dhurandhar et al., and Karandish et al. [2,6,15], who also showed a significant increase in the weight of infected animals compared to control animals. A statistical analysis of our results also confirmed a significant difference in weight between the virus-infected and control embryos. However, our histopathological examinations showed no increase in adipocyte number or size and no tissue hypertrophy, but did show accelerated cell proliferation in the infected embryos. The amount of total fat in organs remained at a normal level, but in some cases, steatosis of some organs was also observed. The results of the experiments performed in this study suggest that the mechanism behind them is significantly different from the known effects of HADv-D36. It seems that the virus finds a susceptible environment in the embryo. The histopathological analyses were consistent with the weight measurements of the embryos.

In the context of organisms’ development, applicable not only to avian species but universally, phases of heightened proliferation exhibit a discernible association with a gradual augmentation in mass. Specifically, certain organs undergo a sequential progression wherein a proliferative phase is succeeded by a distinctly demarcated stage characterized by a reduction in cellular divisions accompanied by hypertrophic processes. This phenomenon is notably exemplified in cells such as muscle or nerve cells, where the predominant proportion of cellular divisions transpires during the initial stages of development. Subsequently, these cells primarily undergo volumetric expansion, elucidating the expeditious alterations observed in the weight of the developing foetus or embryo, particularly during periods of accelerated enlargement in major organ systems. In addition, in embryos that take only 21 days to develop, such growth spikes are natural and often observed [16,17,18,19].

The observed changes might result from an unknown mechanism induced by HAdV-D36 infection. Whether it is due to the high number of differentiating stem cells, or what the mechanism behind it is, still remains unknown. Perhaps it is related to the expression of developmental genes, whose activation suppresses the genes necessary for adipocyte transformation, or perhaps the genes responsible for adipocyte transformation are not yet active.

The only element that seems to work similarly in both adults and embryos is the inflammation caused by the virus. Ha-Na et al. proved that human adenovirus 36 is responsible for increased levels of MCP-1 (macrophage chemoattractant protein 1) because it activates nuclear factor Kß (transcription factor), causing inflammation in adipose tissue [20].

The real-time PCR results showed an unusual fluctuation in the CQ level in the livers of the chicken embryos. The viral load between days 13 and 14 showed a slight increase, but on day 15, it started to decrease, which continued for the next two days. On day 18, the levels of viral DNA increased again. The decrease in the amount of viral genetic material was most likely related to the enhanced cellular response and increased antibody titre [21]. The unusual increase in viral genetic material observed on day 18 was probably connected to the decline in antibody titre, which could no longer stop the virus’s multiplication. However, this hypothesis requires further research, because the small amount of serum in the studied embryos did not allow for the measurement of the level of antibodies. Chicken embryos are characterised by the presence of an immature, but already functioning, immunological system, which comprises antibodies and inflammatory cells [22,23]. The observed fluctuation in CQ levels coincided with the histopathological picture showing a high number of macrophages and lymphocytes between day 13 and day 20 of the embryos’ development.

The heart failure mechanism is associated with changes in fat tissue quantity, inflammatory processes, and altered cardiac physiology, which is additionally complicated by co-morbidities [24]. In these conditions of heart dysfunction, impaired glucose metabolism and fatty acid ß-oxidation disrupt mitochondrial functions and promote the formation of reactive oxygen species (ROS). In the presented study, we observed significantly increased levels of oxidative stress markers—CAT, GPx, TOS, total SOD, CuZnSOD, and MDA—in ECEs’ heart and liver tissue, indicative of increased oxidative stress and ROS levels. Reactive oxygen species negatively affect the function and structure of all cellular macromolecules, including nucleic acids, proteins, and lipids. They can alter Ca^2+^ regulation, activate pathways linked to electrical remodelling, stimulate cardiomyocyte hypertrophy, induce apoptosis, promote fibrosis, and activate or inhibit the inflammatory response. All of these are understood to be crucial factors in the onset of heart failure [25,26]. Reactive oxygen species can modify numerous signalling pathways involved in the hypertrophy of cardiomyocytes. For instance, apoptotic signal-regulating kinase 1 (ASK1) in rat ventricular cardiomyocytes is activated in a redox-dependent manner by angiotensin II, endothelin-1, and phenylephrine, causing their hypertrophy [27]. Our findings indicate that CAT, SOD, GST, and, to some extent, GR activity in the ECEs’ heart tissue is influenced by HAdV-D36 infection, which leads to changes in the oxidative processes and hypertrophy of the heart tissue. In the liver, ROS overwhelm the enzymatic and non-enzymatic antioxidants, causing oxidative stress, hepatocellular dysfunction, and, eventually, hepatic fibrosis. Viral infections also increase the levels of enzymatic and non-enzymatic oxidative stress markers in the liver [28,29]. According to Duygu et al., individuals with chronic HBV infections present higher levels of oxidative stress markers, such as hydroxyl, hydrogen peroxide, singlet oxygen, lipid hydroperoxide, and superoxide, and lower levels of antioxidant markers, such as total sulfhydryl, vitamin C, uric acid, vitamin E, and bilirubin in their blood [30]. Increased glutathione reductase activity prevents the excessive quenching of intracellular ROS essential for insulin signalling [31]. Camini et al. analysed the impact of Caraparu virus (CARV) on hepatic pathogenesis in 6-week-old BALB/c mice and the role of oxidative stress and antioxidant defences in this pathology. CARV infection caused no change in the oxidative stress markers but caused an increase in glutathione content and altered SOD expression and activity [32]. Da Silva et al. studied the impact of Oropouche virus (OROV) on hepatic stress in male and female wild-type BALB/c mice. They reported that SOD and CAT activity in the liver and spleen decreased after OROV infection, indicating the antioxidant response as being ineffective in stopping the oxidative damage [33]. In our study, we observed increased GR and total SOD activities and decreased MnSOD, CuZnSOD, GPx, and GST activities. Moreover, HAdV-D36 infection significantly increased the MDA concentration and TOS levels. The presented data show that oxidative stress generated by HADV-D36 results from an imbalance in redox homeostasis, started by excessive ROS production, that negatively affects biomolecules or incapacitates the antioxidant system.

The results of all the studies carried out in this project indicate that HAdV-D36 has a profound impact on developing foetuses. However, it should be noted that chicken embryos are a very basic research model. Not all observed changes can be directly transferred to more advanced organisms such as human foetuses. In order to extend the knowledge on this topic in the future, it would be necessary to continue the present study using more and more advanced models such as mice or monkeys.

## 4. Materials and Methods

### 4.1. Ethical Statement

The use of embryonated chicken eggs (ECE) was approved by the Local Ethics Committee in Wrocław (approval no. 021/2021; date of approval 17 March 2021).

### 4.2. Virus

A human adenovirus 36 (HAdV-D36) (ATCC VR1610^TM^, Manassas, VA, USA) reference strain, without any modifications, was used in this study.

### 4.3. Embryonated Chicken Eggs (ECEs)

Four hundred and twenty embryonated chicken eggs (ECEs) from Rossa I hens (*Gallus gallus domesticus*) were used in the study. On the fifth day of incubation, ECEs were divided into 4 groups:Ad-AC: inoculated with HAdV-D36 into the allantoic cavity (*n* = 105);Ad-YS: inoculated with HAdV-D36 into the yolk sac (*n* = 105);C-AC: inoculated with PBS into the allantoic cavity (*n* = 105)—control group;C-YS: inoculated with PBS into the yolk sac (*n* = 105 eggs)—control group.

The conventional mode of adenovirus transmission is through respiratory droplets. However, the possibility of HAdV-D36 infection via the oral route cannot be excluded. Therefore, inoculation was performed via the allantoic cavity and yolk sac to imitate both the oral and droplet routes of infection.

### 4.4. Inoculation of ECEs

The edges of the air sacs of the ECEs were marked under an ovoscope. Then, the position of the embryo, the locations of the large blood vessels, and the site of viral inoculation were determined. Two holes in the shell were drilled: above the air sac (to reduce the pressure after the injection) and above the site of inoculation (into the yolk sac or into the allantoic cavity). A volume of 100 µL of 100 TCID_50_ = 10^−6^ with a CQ of 7.23 of adenovirus 36 (or PBS in the case of control embryos) was injected into the yolk sac or allantoic cavity. Then, ECEs were transferred to an incubator and kept at 37 °C and 70% humidity.

### 4.5. ECE Weight Measurements

Starting from day 6 and continuing to day 20 of the incubation period (15 days in total), 28 embryos (7 from each of the 4 groups) were extracted and weighed each day on the laboratory scales (WLCX2, Radwag, Radom, Poland).

### 4.6. Sampling ECEs for Histopathological and Molecular Studies

Chicken embryos removed from the eggs were dissected with a surgical scalpel. Until the 10th day of incubation, the entire embryos were taken due to the small size of the internal organs. After the 10th day of incubation, only livers were sampled for future examinations.

### 4.7. Histopathological Examinations

Whole embryos (up to 10 days) and livers taken from each embryo group were subjected to histopathological examination in order to observe the changes caused by HAdV-D36 infection in the tissues of the examined organism.

The sampled material was fixed in 4% solution of buffered formaldehyde (pH 7.2–7.4). Then, the material was rinsed in tap water and dehydrated in an alcohol series. Eventually, it was embedded in paraffin, sliced into 6 µm thick samples, and stained with haematoxylin and eosin. The material was observed under a light microscope (Nikon Eclipse 80i, Nikon, Tokyo, Japan).

### 4.8. Real-Time PCR

Real-time PCR tests were performed to confirm replication of the virus and to determine the viral load in the samples of liver tissue collected for testing. From each embryo-derived liver, 100mg of tissue was collected for isolation. A commercial Bead-Beat Micro AX Gravity kit (A&A Biotechnology, Gdańsk, Poland) was used to isolate DNA from the samples. Quantitative polymerase chain reaction (qPCR) was performed using an Amplifyme Probe no-rox mix (am04) (Blirt, Gdańsk, Poland) in accordance with the manufacturer’s instructions.

The experimental group comprised isolates derived from the hepatic tissues of embryos infected with HAdV-D36, whereas the control group encompassed isolates obtained from hepatic tissues of embryos that remained uninfected.

Forward primer: AAAGAGCAGCACAGAGAGATCA (Position 34,417–34,438);Reverse primer: GAGTGAGCGTGCTGGTTC (Position 34,533–34,550);Probe: FAM-TTCAAGGCCATAAATCTGCCCTGATATCCA-BHQ1 (Position 34,499–34,528).

The qPCR followed the protocol of 2 min at 50 °C, 10 min at 95 °C, 45 cycles of 20 s at 95 °C, and 1 min at 60 °C.

### 4.9. Antioxidant and Lipid Peroxidation Markers

On the 18th day of ECE development, the heart and liver of HAdV-D36-infected (Ad-AC *n* = 9, Ad-YS *n* = 9) and control (C-AC *n* = 8, C-YS *n* = 8) embryos were collected for antioxidant and lipid peroxidation marker analyses. The samples were homogenized, centrifuged at 12,000× *g* at 4 °C for 10 min, and then stored at −80 °C until the analyses. The analyses were performed using colorimetric methods, and the changes in absorbance were read on a PERKIN ELMER Victor X3 reader (PerkinElmer, Inc., Waltham, MA, USA). The protein content was measured using the Lowry method [34].

#### 4.9.1. Catalase (CAT) Activity (EC 1.11.1.6)

CAT activity was assessed using the Aebi method [35]. CAT activity was expressed as units of activity per 1 g of protein (IU/g protein).

#### 4.9.2. Superoxide Dismutase (SOD) Activity (EC 1.15.1.1)

SOD activity was determined with the Oyanagui method [36]. SOD activity was expressed as nitrite units (NU) per 1 mg of protein, with 1 NU equal to 50% inhibition of nitrite ion formation under the method’s conditions. CuZnSOD activity was assessed by calculating the difference between the total SOD and MnSOD activity in the presence of potassium cyanide (KCN) as a CuZnSOD inhibitor.

#### 4.9.3. Glutathione Peroxidase (GPx) Activity (EC 1.11.1.9)

GPx activity was measured using the kinetic method [37] with t-butyl peroxide as a substrate and expressed as μmoles of NADPH oxidized in 1 min per 1 g of protein.

#### 4.9.4. Glutathione Reductase (GR) Activity (EC 1.8.1.7)

GR activity was assessed using the kinetic method and expressed as μmoles of nicotinamide adenine dinucleotide phosphate (NADPH) utilized in the reaction with oxidized glutathione over 1 min per 1 g of protein [38].

#### 4.9.5. Glutathione-S Transferase (GST) Activity (EC 2.5.1.18)

GST activity was estimated using the Habig and Jakoby kinetic method [39] and expressed as μmoles of thioether formed over 1 min per 1 g of protein.

#### 4.9.6. Total Oxidant Status (TOS)

Total oxidant status (TOS) was determined using the Erel method [40,41].

#### 4.9.7. Malondialdehyde (MDA) Concentration

The malondialdehyde (MDA) concentration was measured using the method of Ohkawa et al. [42] involving the reaction with thiobarbituric acid. The MDA concentration in plasma was calculated against a standard curve prepared from 1,1,3,3-tetraethoxypropane and was expressed in μmol/L.

### 4.10. Statistical Analysis

The normality of the distribution of ECEs’ weight and the antioxidant and lipid peroxidation marker levels was assessed using the Shapiro–Wilk test and a quantile–quantile plot. In the case of ECEs’ weight data, the relationship between quantitative variables was determined using polynomial regression or ordinary least square regression, and data are presented as the mean and 95% confidence interval (95% CI). Data of antioxidant and lipid peroxidation markers with a normal distribution are presented as the mean value and standard deviation (M ± SD), while data with a non-normal distribution are presented as the median and lower and upper quartiles (Me(Q1–Q3)). Student’s *t* test for independent samples was used for group comparisons, and homogeneity of variance was assessed using the Fisher–Snedecor test. The Welch correction test was used in the absence of homogeneity of variance. Differences between groups were determined based on mean values and a 95% confidence interval (CI). For variables deviating from normality, the Mann–Whitney U test was used. All tests were two-tailed, and the significance level was <0.05. Statistical analyses were performed using Statistica v. 13.3.0 [43].

## 5. Conclusions

This article represents an initial stride towards a more exhaustive examination of the impact of adenovirus 36 on the developing organism. Infection of chicken embryos with HAdV-D36 induced a statistically significant increase in weight relative to controls, but unlike in adult models, this was due not to an increase in tissue volume but to increased organogenesis. Our investigation posits the presence of novel and hitherto undescribed mechanisms of this virus’s influence on developing organisms. This newfound understanding of the biological effects of human adenovirus 36 may engender a novel approach to future investigations employing this virus.

## Figures and Tables

**Figure 1 ijms-25-02341-f001:**
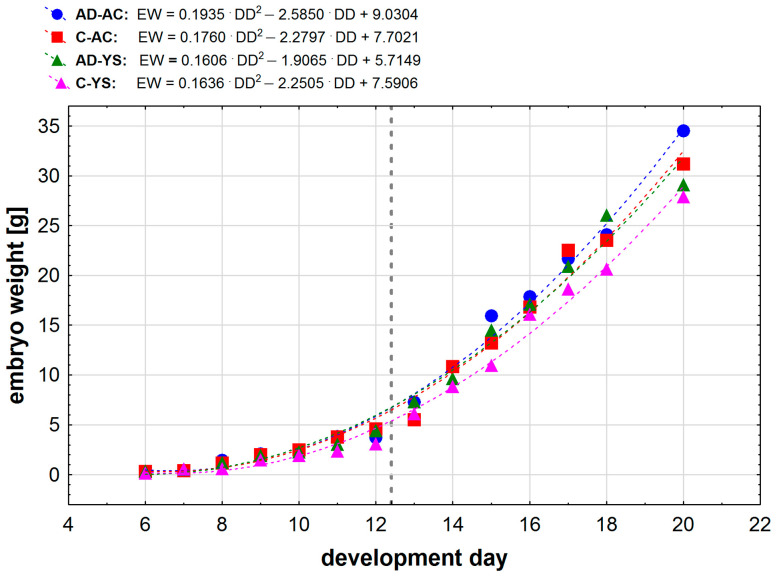
Mean embryo weight in relation to the development day of embryonated chicken eggs (ECEs) inoculated with human adenovirus 36 (HadV-D36) or PBS (control) into the allantoic cavity (AC) or yolk sac (YC) over 15 days of incubation. In the equations, the notation EW means embryo weight [g], and DD means development day. The grey vertical dashed line marks the moment of change in weight gain for all study groups.

**Figure 2 ijms-25-02341-f002:**
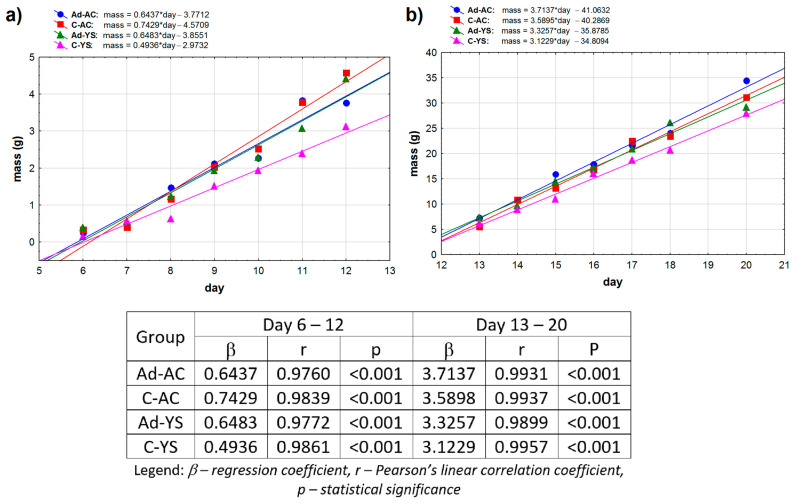
Dependence of mean body weight on age (in days) in the ranges up to day 12 (**a**) and after day 13 (**b**). The linear matches are marked with solid lines.

**Figure 3 ijms-25-02341-f003:**
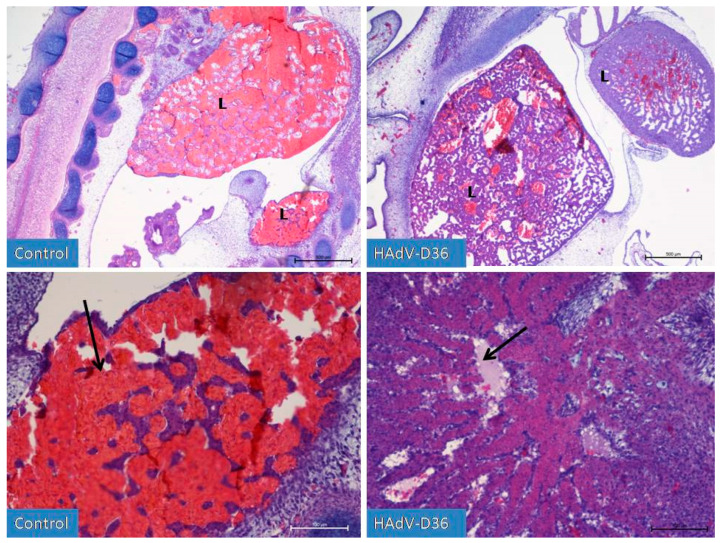
Liver development (L) on the 9th day of incubation of embryonated chicken eggs (ECEs) inoculated with human adenovirus 36 (HAdV-D36) or PBS (control) into the yolk sac. The liver of control ECEs is dominated by wide blood vessels filled with blood (black arrow). In the HAdV-D36-infected ECEs, hepatocytes comprise a significant part of the liver parenchyma, the blood vessels are much less filled with blood, and lymphatic vessels are clearly visible. H&E, Mag. 100, scale bar—500 μm (upper line) and 200×, scale bar—100 μm (lower line).

**Figure 4 ijms-25-02341-f004:**
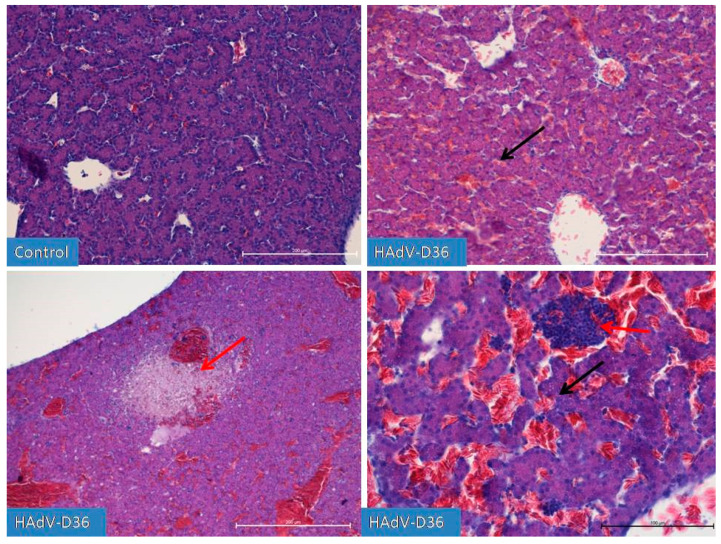
Comparison of the degree of liver development on the 15th day of incubation. The hepatic trabeculae and blood vessels, filled with blood to varying degrees, are clearly visible in both the control and HAdV-D36-infected (inoculated into the yolk sac) ECEs. In the HAdV-D36-infected ECEs, clusters of lymphoid tissue appear in the liver parenchyma, causing local inflammation (red arrow) followed by hepatocyte dystrophy. Fat droplets (black arrow) appear in the hepatocytes. H&E, Mag. 100, scale bar—200 μm and 200×, scale bar—100 μm (bottom-right picture).

**Figure 5 ijms-25-02341-f005:**
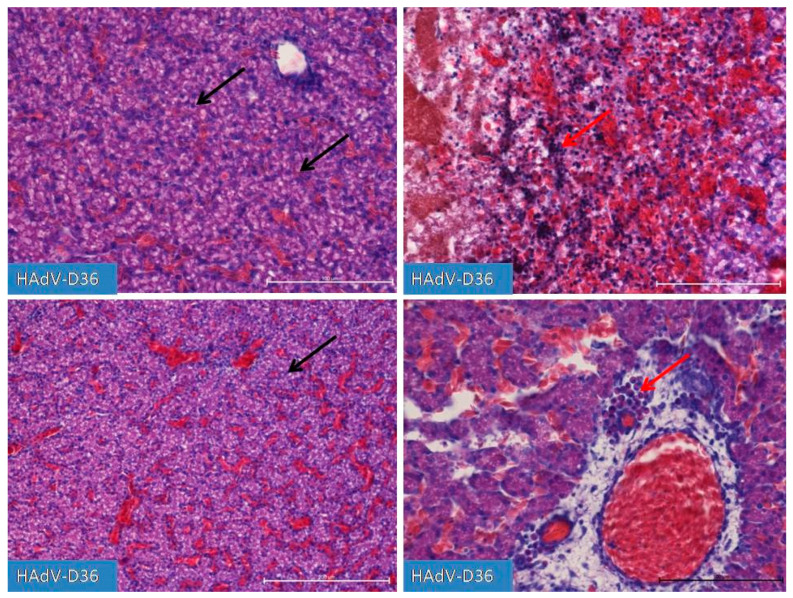
Liver after infection with HAdV-D36 into the yolk sac on day 19 of incubation. Increasing lipogenesis occurs in hepatocytes (black arrow), accompanied by macrophage and lymphocytic infiltration (red arrow) in the vicinity of blood vessels and in the liver parenchyma. The increased presence of congestive blood is strongly diverse and region-dependent. Changes in the liver parenchyma are accompanied by the degradation of hepatocytes. H&E, Mag. 100, scale bar—100 μm and 200×, scale bar—200 μm (bottom-right line).

**Figure 6 ijms-25-02341-f006:**
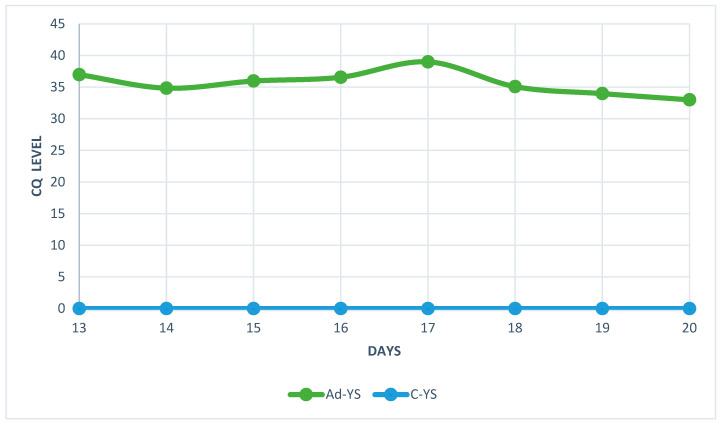
Quantification cycle (CQ) level of HAdV-D36 genetic material isolated from the livers of embryonated chicken eggs (ECEs) inoculated with human adenovirus 36 or PBS into the yolk sac.

**Figure 7 ijms-25-02341-f007:**
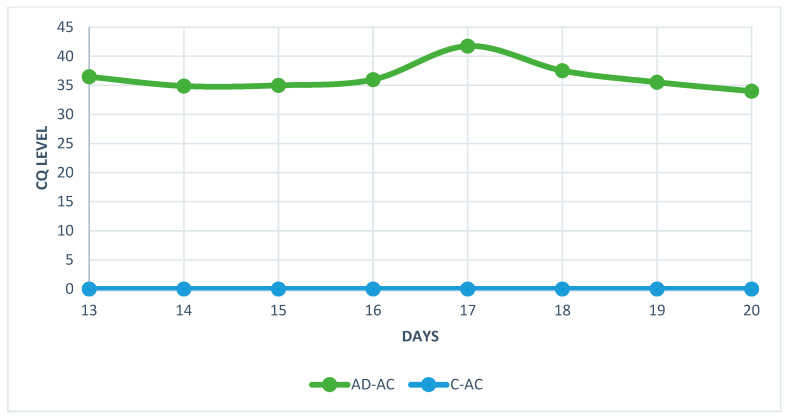
Quantification cycle (CQ) level of HAdV-D36 genetic material isolated from the livers of embryonated chicken eggs (ECEs) inoculated with human adenovirus 36 or PBS into the allantoic cavity.

**Table 1 ijms-25-02341-t001:** Weights of embryos from embryonated chicken eggs (ECEs) inoculated with human adenovirus 36 (HAdV-D36) or PBS (control) into the allantoic cavity (AC) or yolk sac (YC) over 15 days of incubation. Results are presented as the mean ± standard deviation (Me ± SD).

Day of Incubation		ECEs Weight [g]				
	Ad-YS	C-YS	% of Difference	Ad-AC	C-AC	% of Difference
6	0.38 ± 0.12	0.15 ± 0.04	50%	0.24 ± 0.10	0.34 ± 0.02	30%
7	0.55 ± 0.17	0.57 ± 0.01	3%	0.47 ± 0.14	0.40 ± 0.11	17%
8	1.23 ± 0.38	0.63 ± 0.08	95%	1.47 ± 0.51	1.16 ± 0.26	26%
9	1.93 ± 0.39	1.50 ± 0.38	28%	2.15 ± 0.21	2.03 ± 0.43	5%
10	2.28 ± 0.43	1.93 ± 0.13	18%	2.26 ± 0.45	2.51 ± 0.47	10%
11	3.06 ± 0.65	2.39 ± 0.23	28%	3.82 ± 0.90	3.78 ± 0.33	1%
12	4.41 ± 1.32	3.12 ± 0.54	41%	3.78 ± 1.00	4.58 ± 0.15	18%
13	7.33 ± 2.04	6.13 ± 0.83	19%	7.30 ± 1.44	5.52 ± 0.12	32%
14	9.64 ± 2.37	8.84 ± 2.78	9%	10.86 ± 0.46	10.86 ± 2.00	0%
15	14.48 ± 2.25	10.97 ± 2.12	31%	15.93 ± 0.53	13.20 ± 1.60	20%
16	17.12 ± 4.29	16.12 ± 1.51	6%	17.87 ± 0.33	16.82 ± 0.62	6%
17	20.91 ± 0.65	18.62 ± 2.91	12%	21.66 ± 0.50	22.63 ± 3.62	5%
18	26.04 ± 1.13	20.63 ± 2.65	26%	24.08 ± 0.39	23.52 ± 3.62	2%
19	27.59 ± 2.01	26.45 ± 2.04	4%	26.19 ± 0.19	27.66 ± 0.98	6%
20	29.13 ± 2.32	27.91 ± 1.83	4%	34.51 ± 0.30	31.18 ± 0.18	10%

**Table 2 ijms-25-02341-t002:** Antioxidant and lipid peroxidation markers’ levels in the heart tissue of embryonated chicken eggs (ECEs) inoculated with human adenovirus 36.

HEART	CONTROL *n* = 16	HAdV-D36-Infected *n* = 18	t/z *	*p*
CAT (IU/g protein)	1584 ± 205	2585 ± 798	5.13	<0.001
GR (IU/g protein)	2.6 (2.3–3.0)	3.0 (2.2–3.9)	0.57 *	0.569
TOS (μmol/g protein)	2.6 ± 0.5	2.5 ± 0.8	0.22	0.826
SOD (NU/mg protein)	58.0 (52.5–59.0)	74.6 (70.9–76.2)	4.47 *	<0.001
MnSOD (NU/mg protein)	33.8 ± 4.1	40.8 ± 8.0	2.41	<0.05
CuZnSOD (NU/mg protein)	22.4 ± 5.1	31.6 ± 12.9	2.81	<0.05
GPx (IU/g protein)	4.0 ± 0.9	3.8 ± 0.9	0.62	0.542
GST (IU/g protein)	1.24 ± 0.23	1.62 ± 0.49	2.19	<0.05
MDA (μmol/g protein)	1.6 (1.2–1.8)	3.3(1.8–5.1)	3.05 *	<0.01

Legend: CAT (IUiu/g protein)—catalase, CuZnSOD (NU/mg protein)—copper/zinc superoxide dismutase, GPx (IU/g protein)—glutathione peroxidase, GR (IU/g protein)—glutathione reductase, GST (IU/g protein)—glutathione S transferase, MDA (μmol/g protein)—malondialdehyde, MnSOD (NU/mg protein)—manganese superoxide dismutase, SOD (NU/mg protein)—superoxide dismutase, TOS (μmol/g protein)—total oxidative status, *t*—test statistic in the *t*-Student test for independent samples, z *—test statistic in the Mann–Whitney test, *p*—statistical significance, M ± SD—mean ± standard deviation, Me(Q_1_–Q_3_)—median (lower–upper quartiles).

**Table 3 ijms-25-02341-t003:** Antioxidant and lipid peroxidation markers’ levels in the liver tissue of embryonated chicken eggs (ECEs) inoculated with human adenovirus 36.

	Control *n* = 16	HAdV-D36-Infected *n* = 18	t/z *	*p*
CAT (IU/g protein)	4596 ± 1255	4834 ± 1487	0.50	0.617
GR (IU/g protein)	0.84 ± 0.15	1.31 ± 0.37	5.04	<0.001
TOS (μmol/g protein)	2.1 ± 0.5	2.9 ± 0.9	2.93	<0.01
SOD (NU/mg protein)	63.3 ± 7.9	90.4 ± 30.6	3.30	<0.01
MnSOD (NU/mg protein)	40.1 ± 7.1	38.0 ± 16.8	0.50	0.623
CuZnSOD (NU/mg protein)	28.0 (18.8–31.2)	38.0 (31.9–66.7)	3.36 *	<0.001
GPx (IU/g protein)	7.0 ± 1.3	5.5 ± 1.3	3.58	<0.01
GST (IU/g protein)	5.14 ± 0.99	4.72 ± 1.14	1.15	0.258
MDA (μmol/g protein)	0.77 (0.60–1.08)	1.35(1.16–1.92)	2.43 *	<0.05

Legend: CAT (IUiu/g protein)—catalase, CuZnSOD (NU/mg protein)—copper/zinc superoxide dismutase, GPx (IU/g protein)—glutathione peroxidase, GR (IU/g protein)—glutathione reductase, GST (IU/g protein)—glutathione S transferase, MDA (μmol/g protein)—malondialdehyde, MnSOD (NU/mg protein)—manganese superoxide dismutase, SOD (NU/mg protein)—superoxide dismutase, TOS (μmol/g protein)—total oxidative status, *t*—test statistic in the *t*-Student test for independent samples, z *—test statistic in the Mann–Whitney test, *p*—statistical significance, M ± SD—mean ± standard deviation, Me(Q_1_–Q_3_)—median (lower–upper quartiles).

## Data Availability

The datasets used and/or analysed during the current study are available from the corresponding author on reasonable request.

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
