# Peer review of "Impact of Human Adenovirus 36 on Embryonated Chicken Eggs: Insights into Growth Mechanisms"

_ijms, 2024, doi:10.3390/ijms25042341_

Round 1
Reviewer 1 Report
Comments and Suggestions for Authors
IJMS-2809006-Peer-Review-Report-v1
The manuscript is well-written and current and relevant materials were consulted.
Title Refinement:
The title is engaging and informative but can be improved. Suggestion: Exploring the Impact of Human Adenovirus 36 on Embryonated Chicken Eggs: Insights into Growth Mechanisms.
Your abstract provides valuable information on the effects of Human adenovirus 36 (HAdV-D36) on embryonated chicken eggs (ECEs). It appears too concise; it should be reviewed to contain some more details, especially the results of your study. However, to make it more attractive to readers, consider the following improvements:
1. Clearly state the objective of your study. For example, "This study aims to investigate the impact of HAdV-D36 on embryonic development by utilizing embryonated chicken eggs as a dynamic model."
2. The term "infectious agent capable of promoting obesity" might be misleading. It would be more accurate to state that HAdV-D36 has been associated with an increased risk of obesity rather than implying a direct promotion.
3. The statement "HAdV-D36 must affect the developing organisms via another mechanism inducing enhanced ECEs growth" is somewhat vague. The abstract could benefit from further elucidation of the proposed mechanism, providing insights into how HAdV-D36 influences cell proliferation in the absence of adipose tissue development.
4. Although the abstract states a comparison with controls, it lacks details on the control group and how the comparison was conducted. Providing more evidence on the controls and statistical methods used for comparison would reinforce the study.
5. Some sentences could be rephrased for better understanding. For instance, "but increased cell proliferation of all tissues" could be improved to offer a more specific description of the affected tissues.
It is important for the authors to conclude the discussion by summarizing the significant findings and their implications for the broader field. Provide a concise summary of the study's major contributions. However, the conclusion provides closure to the reader, highlighting the key takeaways and emphasizing the study's contribution to the existing body of knowledge. Additionally, it may highlight the limitations of the research and suggest areas for further research. However, it is optional in this journal, but based on the reasons highlighted above, the authors may leave the manuscript as it is or include conclusions as a subheading.
Lines 288-290: For clarity and correctness, the sentence should be: Our measurements of the chicken embryos weight changes agree with the observations made by Pasarica et al., Dhurandhar et al., and Karandish et al. [2,6,22], who also showed a significant increase in the weight of infected animals compared to control animals.
Line 357: Camini et al was cited in the text but not on the list of references.
Line 475/476: Recast the sentence for clarity. Suggestion: The Welch correction test was used in the absence of homogeneity of variance.
Line 476: Change group to its plural form.
The materials and methods section outlines the experimental techniques performed on embryonated chicken eggs (ECEs) to explore the results of Human Adenovirus 36 (HAdV-D36) infection. In general, the segment is comprehensive, but there are a few points to consider by the authors:
1. Indicate whether any modifications were made to the ATCC VR1610TM HAdV-D36 strain.
2. Clarify the rationale for deciding the allantoic cavity and yolk sac for inoculation to imitate oral and droplet routes of infection.
3. Provide details on the choice of the injection site (yolk sac or allantoic cavity) and how this choice may impact the study.
4. Ensure that the injection volume of 100 μl is justified and explain its importance. Any variations in the injected volume should be mentioned.
5. Specify why the conditions of 37 °C and 80% humidity were chosen for incubation. Give references or explanations if applicable.
6. Clarify your justifications for extracting and weighing embryos daily from day 6 to day 20. Explain how this daily monitoring impacts the objectives of the study.
7. Give additional details on the criteria used for selecting embryos for histopathological and molecular studies.
8. Justify the significance of choosing whole embryos up to the 10th day and only livers after the 10th day.
The statistical analysis approach described in the study is appropriate, but the authors should be mindful of the following:
1. Software Version and Citation: The usage of a specific software version (Statistica v. 13.3.0) is stated in the text but not in the reference section. Proper citation of the software is essential for transparency and reproducibility.
2. The t-test believes that the data are normally distributed, and variances are equal between the groups. The handling of the Welch correction is suitable for unequal variances, but it's important to check if the assumptions are met.
3. Were there any data transformations or handling of outliers? It's essential to investigate the influence of outliers on the outcomes and weigh up transformations if needed.
The authors used current literature to write the paper. However, they did not strictly follow the MDPI guidelines on referencing, especially in punctuation.

Requires minor editing.
Reviewer 2 Report
Comments and Suggestions for Authors
Pogorzelska and coauthors report findings on the effect of human adenovirus 36 (which by convention should be written HAdV-D36) infection on the growth and development of embryonated chicken eggs (ECE). The study is of interest because HAdV-D36 has been shown to be adipogenic in cell culture and animal models and has been correlated in most studies with obesity in humans. However, there are a number of issues that should be addressed as pinpointed below.
1. The abstract and introduction should be revised and modified substantially to avoid a series of imprecisions (some of them pinpointed below), in particular stating the unfounded conclusion that HAdV-D36 is known to cause or promote obesity in humans. Most studies attempting to determine whether HAdV-D36 infection may be associated with obesity in humans have correlated the presence of serum antibodies vs HAdV-D36 with the Body Mass Index (BMI) or other blood biochemical parameters. Only a few studies have attempted to determine the presence of the virus in humans and no study has determined that the virus is an etiologic agent for obesity in humans. This should be considered throughout the ms and corrected where appropriate.
2. L52-54: Cite the articles that have shown reduction in blood triglycerides and cholesterol, and those that have shown an adipogenic effect in animals or cell culture separately. A direct effect of the virus infection on adipogenesis in humans has not been determined.
3. L63-64: The sentence, "The worldwide studies on various populations show the presence of adenovirus 36 in the majority of the studied subjects." is not correct. Only a few studies have reported the presence of HAdV-D36 in adipose or any other tissue in humans. As indicated above, most studies have correlated HAdV-D36 seropositivity with BMI, not the presence of the virus.
4. L68-70: As noted above, the role of HAdV-D36 in obesity in humans has only been studied using correlative studies and it has not been demonstrated that the virus is an etiologic agent of obesity. Therefore, the sentence: "Similar studies conducted worldwide present similar results indicating the increasing role of HAdV-D36 69 in the incidence of infectious obesity." is not correct.
5. L71-72: As above, the mechanism through which the virus exerts an adipogenic effect has only been studied in cell culture. Therefore, the sentence "So far, the mechanisms of action have been studied and described only in adult patients or adult model organisms." is also incorrect. Moreover, there are a few studies that have attempted to correlate HAdV-D36 seroprevalence and obesity in children.
6. L74-75: The idea behind the sentence " Viruses, being similir to insulin, may possibly act as glucose uptake inductors or growth factors." should be developed and justified. Viruses are not similar to insulin, glucose uptake inductors or growth factors, rather virus infection may have similar effects.
7. L75-79: Why teratogenic? and When/where did screening pregnant women and animals for viruses become standard?
The paper cited by the authors by Anna Chudnovets and coworkers reviews the pathogenesis of viral infections in the context of pregnancy, but do not mention that viruses have a teratogenic effect (they cite the association of the use of ribavirin with a teratogenic effect), nor mention that screening for viruses in pregnant women and animals has become standard.
8. L79-80: For the reasons noted above, it cannot be said that "the virus affects the adults via complex mechanism". The mechanism exerted by viral infection in humans is not known and can only be inferred from cell culture or animal model studies.
9. L48-49: The description of the viral genome stability cannot be expressed in terms of the years it took to be determined.
The paper cited by the authors by J-H Nam and coworkers reported a comparison between the whole genome sequence of a HAdV-D36 virus obtained from the ATCC in 1988, with the genome - sequenced 14 years later - of a virus that was passaged "approximately" 12 times. Although these passages were done during a period of 14 years, the time is not of relevance, but rather the number of passages. In the study by J-H Nam the authors report a genetic variation of ∼2.37 × 10–6 mutations /nucleotide /passage, which is a more appropriate term to describe the stability of the viral genome.
10. L79-82: Please review the writing of the sentence. And what do the authors mean by "a lot of new knowledge"?
Also, I suggest to better justify the use of ECEs as a model to study the effect of HAdV-D36 infection in developing embryos. Surely, it should be interesting to know why the authors chose to use the ECEs for their study.
Results.
Embryo weight.
11. I suggest the authors should explain briefly why inoculation with the virus of the allantoic cavity and the yolk sack of ECEs can be expected to simulate the oral and "droplet" route of infection.
12. I suggest, the differences in weight at the various time-points between the virus-inoculated and PBS-inoculated embryos may be better described in relative terms (as percentage) instead of number of grams.
13. Although statistically significant the differences in weight seem rather small. Are these differences comparable to other studies that use ECEs to determine the effect of an infectious agent? If so, the authors should discuss this and include the reference(s). For example: The largest numerical difference in Table 1 corresponds to a 26.2% difference between the Ad-YS and C-YS at the 18th day of incubation, which seems substantial; however, at the 19th day (presumably 24 hours later) the difference is only 4.3%. Does a 26% rise in weight and a 20% decline in 24 hours make any physiological sense?
14. In Figure 1. why is the grey dashed line considered to mark "the moment" of change? What could be the physiological meaning if any of that time-point?
Histopathological examination.
15. I suggest the authors use a more systematic description of their data and explain in more detail their observations referring each observation to the images shown in the figures. As it stands this section often appears anecdotic. The authors should describe how many samples were analyzed in each case and report in what percentage of the analyzed samples each of the described observations were made.
Viral DNA.
16. The authors should describe what they mean by "100 µl of 100 TCID50 = 10-6 of virus". As stated it makes no sense. How many infectious units were used to inoculate the ECEs? What virus was used and how was the virus amplified and titered?
17. The method for viral DNA quantification should be described in detail. Did the authors use absolute or relative quantification? If relative, what was used as reference? What are CQ and Cq values?
Determination of viral DNA replication requires a comparison with the input that was used to initiate the infection, therefore, Figs 5 and 6. should include determination of viral DNA from the initial time-point. How did the authors determine viral replication or viral load?
Also, if, as the authors claim HAdV-D36 infection induces cell proliferation, what were the numbers of cells used in each case to obtain and measure viral DNA? Were the cell numbers normalized?
18. What do the authors mean by "Eorf4 gene"? Do they refer to E4orf1 or to the E4 region of the viral genome?
Antioxidants and lipid peroxidation
19. Although the data in this section are clear, the authors should attempt to justify what was the purpose of these experiments, as they seem rather disconnected from the previous experiments.
Discussion.
The discussion is excessively lengthy and includes multiple speculations that should be avoided. I suggest the authors discuss some of the points raised above in order to attempt to interpret their results in the context of what is known about the effect of HAdV-D36 infection on metabolic reprogramming of the cell and on cell proliferation.
20. L283: The sentence "The influence of the virus on the host organism are comprehensively described" is not justified and should be avoided. There are many aspects of the biology of the virus that are incompletely understood and even more of the effect the infection may have on the host. What exactly is meant by "the influence of the virus"?
18. L284-285: In line with the comments made above, the effect of HAdV-D36-infection in humans has not been determined. However, the many correlation studies have included not only adults but also in some cases children.
19. L286-287: In the sentence "Our data suggest that HADV-D36 effect on embryonated chicken eggs (ECEs) results from completely different mechanism than the one described by Poterio et al. [3]." the authors refer a review by Ponterio et al, what mechanism do the authors refer to in this review?
20. L290-291: "Also, statistical analysis of our results confirms significant difference in the weight between the virus-infected and control embryos." However, this is not the case at every time-point. The authors should discuss this in the context of point 13. above.
Other points:
L47 & 52: By convention virus family names should be italicized: Adenoviridae
L48: Should read: "It is a non-enveloped virus with stable, highly conserved genome..." Not "conservative".
L49-50: Please review the writing of the sentence: This virus causes mainly the respiratory and the digestive system diseases.
L55: Please review: "The virus affects the differentiation of 3T3-L1 preadipocytes into adipocytes"
L89 & L137: Replace "these" with "those".
Comments on the Quality of English LanguageThe manuscript is very difficult to follow. I suggest comprehensive review and editing to improve clarity.
Round 2
Reviewer 1 Report
Comments and Suggestions for Authors
IJMS-2809006-Peer-Review-Report-v2
I am happy with the quality of the corrections and implementation of suggestions from the initial review. The authors did a great job of addressing these. This has greatly improved the paper from the form it was presented. Your concluding section, which was added as a suggestion from the initial review, provides a concise summary of the study and suggests potential implications for future research. However, there are a few areas where you might consider making improvements:
a. Consider summarizing key findings or results in the conclusion. If there are quantitative results or statistical significance, briefly mention them to reinforce the study's impact.
b. Expand on the potential practical implications of the findings. How might this newfound understanding of HAdV-D36's effects influence future medical research or applications? Addressing this can add depth to the conclusion.
c. Consider briefly acknowledging any limitations of the study and suggesting directions for future research. This helps situate your work within the broader scientific context and demonstrates a thoughtful consideration of the study's implications.

Author Response
Dear Reviewer,
Thank you very much for all your valuable comments allowing us to improve the manuscript we have presented. We hope that the corrections made in accordance with your suggestions will significantly improve the level of the text and positively influence its reception.
I am happy with the quality of the corrections and implementation of suggestions from the initial review. The authors did a great job of addressing these. This has greatly improved the paper from the form it was presented. Your concluding section, which was added as a suggestion from the initial review, provides a concise summary of the study and suggests potential implications for future research. However, there are a few areas where you might consider making improvements:
- Consider summarizing key findings or results in the conclusion. If there are quantitative results or statistical significance, briefly mention them to reinforce the study's impact.
The proper sentences have been added to conclusion section
- Expand on the potential practical implications of the findings. How might this newfound understanding of HAdV-D36's effects influence future medical research or applications? Addressing this can add depth to the conclusion.
Knowing the exact mechanisms by which adenovirus 36 may accelerate fetal organogenesis could be of significant importance for both medicine and veterinary medicine. This could explain the phenomenon of unexpectedly high birth weights observed in neonatology, as well as explain liver inflammation in neonates. However, these are such far-fetched conjectures that for the time being we dare not mention them in the publication. We plan to conduct similar studies in mammals to confirm this hypothesis.
- Consider briefly acknowledging any limitations of the study and suggesting directions for future research. This helps situate your work within the broader scientific context and demonstrates a thoughtful consideration of the study's implications.
The proper sentence has been added to the discussion section:
However, it should be noted that chicken embryos are a very basic research model. Not all observed changes can be directly transferred to more advanced organisms such as human foetuses. In order to extend the knowledge on this topic in the future, it would be necessary to continue the present study using more and more advanced models such as mice or monkeys
Reviewer 2 Report
Comments and Suggestions for Authors
The modified title does not seem appropriate: the experiments do not provide insights into mechanisms of growth.
L68: Replace, "Adenoviruses causes" by "Adenoviruses cause"
L72: Replace, "the number of triglycerides and cholesterol" by "the amount of triglycerides and cholesterol"
L67: Correct the number in the ms from 2.37 106 to 2.37 10-6
L82: I suggest replacing, "The worldwide studies on various populations show the presence of antibodies adenovirus 36 in the majority of the studied subjects." by "The worldwide studies on various populations show the presence of antibodies against adenovirus 36 in the majority of the studied subjects."
L87-88: I suggest replacing, "Similar studies conducted worldwide present similar results indicating the increasing potential role of HAdV-D36 involvement in causing infectious obesity." by "Similar studies conducted worldwide present similar results indicating the increasing potential role of HAdV-D36 involvement in the incidence of infectious obesity."
To avoid sensationalism:
L101: I suggest to find an alternative to: "...a lot of new knowledge..."
L105-106: I suggest to find an alternative to: "...render this research model remarkably versatile and eminently well-suited..."
13. Although statistically significant the differences in weight seem rather small. Are these differences comparable to other studies that use ECEs to determine the effect of an infectious agent? If so, the authors should discuss this and include the reference(s). For example: The largest numerical difference in Table 1 corresponds to a 26.2% difference between the Ad-YS and C-YS at the 18th day of incubation, which seems substantial; however, at the 19th day (presumably 24 hours later) the difference is only 4.3%. Does a 26% rise in weight and a 20% decline in 24 hours make any physiological sense?
An explanation of this issue is included in the text of the manuscript
I could not find a clear the explanation in the modified ms.
Histopathological examination.
15. I suggest the authors use a more systematic description of their data and explain in more detail their observations referring each observation to the images shown in the figures. As it stands this section often appears anecdotic. The authors should describe how many samples were analyzed in each case and report in what percentage of the analyzed samples each of the described observations were made.
The precise quantity of animals employed in the experimentation is delineated in the Materials and Methods section. No statistically significant distinctions were discerned among the animals comprising the observed cohort. The sole factor exhibiting a marginal disparity between the groups was the embryo mortality rate, which exhibited an elevated incidence in the control group; however, it is imperative to interpret this occurrence as stochastic in nature.
This point should be discussed in the ms.
Also, since no cell count can be obtained, then no cell count was obtained for adipocytes and the conclusion in L226 ("No increased number of adipocytes in the chicken body was observed.") is not substantiated by the experiments or interpretation.
Viral DNA.
17. The method for viral DNA quantification should be described in detail. Did the authors use absolute or relative quantification? If relative, what was used as reference? What are CQ and Cq values?
The method of DVA quantification is described in detail in the manufacturer manual of used kits. In the study, relative quantification was performed, where the test sample was liver tissue from virus-infected embryos and the control was tissue from uninfected embryos. For clarification of this matter the corresponding sentence has been added in the text.
CQ and Cq are the same values it’s writing mistake which was corrected in manuscript.
L283: Not corrected.
Determination of viral DNA replication requires a comparison with the input that was used to initiate the infection, therefore, Figs 5 and 6. should include determination of viral DNA from the initial time-point. How did the authors determine viral replication or viral load?
Viral DNA could not be determined from the time of administration because embryonic liver tissue was used for the PCR reaction. The liver up to day 13 of development is too small for the amount of test material to be sufficient for both PCR and histological preparations. Only from day 13 onwards was the amount of material sufficient. With a view to good practice in working with model organisms and minimizing animal suffering, we decided to carry out assays only from the point in development when there was sufficient tissue amount.
The information about initial amount of virus used in the test has been included in the materials and methods section
There is no need to quantify the amount of viral DNA in the inoculated embryo at the time of administration, but the CQ of the viral DNA used to inoculate the ECEs should be quantified in order to determine whether viral DNA increased.
Also, if, as the authors claim HAdV-D36 infection induces cell proliferation, what were the numbers of cells used in each case to obtain and measure viral DNA? Were the cell numbers normalized?
As described in the materials and methods section, qPCR studies were performed using liver tissue cells. A quantity of 100mg of tissue was taken for each test; it was impossible to calculate the number of cells. As the sentence describing the test material may be misleading, it has been amended in the materials and methods section.
Due to the nature of liver growth, we do not know of an effective method to normalise the number of hepatocyte cells extracted from the liver, so we decided to use tissue weight rather than cell count, knowing that differences due to the stage of development, such as water, blood and connective tissue content, will affect the number of liver cells
Is it possible then to reach any conclusion about cell number? See also point 15. above.
Discussion.
The discussion is excessively lengthy and includes multiple speculations that should be avoided. I suggest the authors discuss some of the points raised above in order to attempt to interpret their results in the context of what is known about the effect of HAdV-D36 infection on metabolic reprogramming of the cell and on cell proliferation.
This point has not been addressed. Much is known about the effect of adenoviruses on metabolic reprogramming of the cell and on cell proliferation.
19. L286-287: In the sentence "Our data suggest that HADV-D36 effect on embryonated chicken eggs (ECEs) results from completely different mechanism than the one described by Poterio et al. [3]." the authors refer a review by Ponterio et al, what mechanism do the authors refer to in this review?
The mechanisms to which authors refers has been added to the manuscript text.
The authors' data do not provide evidence to conclude that adipocyte differentiation, heightened cellular glucose uptake, and endocrine disruption are not implicated in their model.
20. L290-291: "Also, statistical analysis of our results confirms significant difference in the weight between the virus-infected and control embryos." However, this is not the case at every time-point. The authors should discuss this in the context of point 13. above.
In the context of organismal development, applicable not only to avian species but universally, phases of heightened proliferation exhibit a discernible association with a gradual augmentation in mass. Specifically, certain organs undergo a sequential progression wherein a proliferative phase is succeeded by a distinctly demarcated stage characterized by a reduction in cellular divisions accompanied by hypertrophic processes. This phenomenon is notably exemplified in cells such as muscle or nerve cells, where the predominant proportion of cellular divisions transpires during the initial stages of development. Subsequently, these cells primarily undergo volumetric expansion, elucidating the expeditious alterations observed in the weight of the developing fetus or embryo, particularly during periods of accelerated enlargement in major organ systems.
In addition, in embryos that take only 21 days to develop, such growth spikes are natural and often observed
This should be briefly discussed
Other points:
L47 & 52: By convention virus family names should be italicized: Adenoviridae
The mistake has been corrected
Not corrected
Comments on the Quality of English Language
The manuscript is very difficult to follow. I suggest comprehensive review and editing to improve clarity.
Language proofreading has been done. Manuscript has been checked for accuracy by Mr Scott Richards, who is a native with a 1 st class Masters Degree in English.
Although the English language is correct in the modified ms, I still suggest to review further the scientific language and clarity in many instances. For example:
L31-34. What do the authors understand by "dynamic model"? "Developing organism" seems a better choice.
L40. I suggest replacing "scrutinized" by "examined" or "analyzed".
L41-42: I suggest replacing "Our investigation substantiates a noteworthy increase in the ECEs body weight of ECEs." with the author's previous text: "Our study confirmed a significant increase in the ECEs' body weight".
L378: Please review: "The influence of the virus on a host organism are..."
L603-608: Please review the writing and substance of the Conclusion. As it stands it seems to be meant to obfuscate the reader.
Does the virus facilitate organogenesis?
Is the virus ostensibly innocuous?
In my view there is no clear scientific meaning in these sentences.
Author Response
Dear Reviewer,
Due to the different colors used in the responses, please take a look at the attached file.
Kind regards
Barbara Bażanów

Round 3
Reviewer 2 Report
Comments and Suggestions for Authors
The authors have addressed all points.